# Economic Role of Government Budget Revision in the Presence of COVID-19

**Ledjon Shahini** [1,*] and **Perseta Grabova** [2,*]

1   Department of Economics, Faculty of Economy, University of Tirana, 1010 Tirana, Albania
2   Department of Finance, Faculty of Economy, University of Tirana, 1010 Tirana, Albania
*   Correspondence: ledjon.shahini@unitir.edu.al (L.S.); perseta.grabova@unitir.edu.al (P.G.)

**Abstract:** Fiscal policies are one of the most important instruments of government to guide the progress of the country's economic development. They find significant use in cases where the economy is experiencing a period of recession, such as the current one caused by COVID-19. This study aims to assess the multiplier effects that budget revision has on the economy for the case of Albania, and more specifically by referring to the initial and revised budget scenario for the year 2020 which is characterized by significant changes caused by the presence of COVID-19. Referring to the multipliers from the input–output tables (IOT) the total effect that the state budget brings to the economy for a certain year is derived. From this paper, it appears that the budget restructuring that takes place during the year does not take into account the multiplier effect in the economy, but is mostly done for specific purposes related to certain government functions. In this context, it is very important that various options during budget revision are evaluated, concluding with the option that has the highest returns for the economy.

**Keywords:** COVID-19; fiscal policy; multiplier; national budget; public expenditures

## 1. Introduction

The COVID-19 pandemic, although having been predicted (Menachery et al. 2015) and despite clear warnings (World Health Organization (WHO) 2011) on the vulnerability of the health system throughout a pandemic, rapidly spread globally with major consequences for public health and the economy. Recognizing the insidious nature of this virus, many governments, to prevent its spread, responded with virus control strategies such as banning gatherings, closing schools, staying at home and so on. All the above measures require social distancing which reduces economic activity while creating high economic costs. If the economic policies are not aggressive enough and continued macroeconomic stimuli are not used, they could make the economic effects of the crisis even larger (Cerra et al. 2020, p. 39).

Referring to economic policies, World Bank (WB) experts (Loayza and Pennings 2020) have suggested in the short term to take mitigation measures for businesses and the most vulnerable groups. In the medium term, experts from the MFI and the WB, in order to limit the negative impact of this crisis on public financial capacity or even households, as well as to ensure a fast recovery after the crisis, suggest the necessity of using monetary, fiscal and financial incentives (Chen et al. 2020; Loayza and Pennings 2020). However, it is difficult to determine the definition of policy that can have a greater impact on the recovery process. Among the fiscal incentives that can be used to combat economic downturn we can mention tax cuts, incentives in direct spending and various investment programs.

Currently, one of the most controversial macroeconomic issues both theoretically and empirically in the world is the impact that fiscal policies have on total gross output and the size of fiscal multipliers. If we refer to the existing theoretical frameworks, we note that they provide inconsistent results on the impact that public spending has on

production (neoclassical theories vs. Keynesian theories). Ambiguous results are also found in empirical studies. On one side there are numerous studies that prove that the highest impact of fiscal policy has been observed in periods of recession (Auerbach and Gorodnichenko 2012) and on the other side other authors (Ramey and Zubairy 2018; Owyang et al. 2013) show that no evidence of significant multiplier changes in the economy in the economic downturn situation is found. Furthermore, from a theoretical point of view, it is not clear whether multipliers should be expected to be higher or lower in emergency economies than in advanced economies (Batini et al. 2014, p. 8). Little consensus also exists in the literature regarding the size of fiscal multipliers. There are numerous studies proving a variance of them from zero to four depending on the different methodologies used or even by countries (Batini et al. 2014; Riera-Crichton et al. 2015). Fiscal multipliers depend on country characteristics such as exchange rate regime, level of public debt, degree of trade openness, savings rates, financial development, level of income, size of the government, composition of spending and taxes, and business cycle. (Ilzetzki et al. 2013; Restrepo 2020). "The multiplier" is a nebulous concept that depends very much on the type of government spending, its persistence, and how it is financed (Ramey 2011, p. 1). The evidence for higher government spending multipliers during periods in which monetary policy is very accommodative, such as zero lower bound periods, is somewhat stronger (Ramey 2019, p. 90). In developing countries in the absence of sufficient fiscal space (small multipliers) the effectiveness of fiscal incentives may be low (Loayza and Pennings 2020).

The difficulty of estimating the multiplier effects of fiscal policies in the country is also due to the fact there is no direct link between government spending and economic activities. The most recent methodological document, "Manual on sources and methods for the compilation of COFOG (Classification of the Functions of Government) statistics" (2019 edition), provides a detailed overview of the relationship between COFOG and the European System of Accounts (Eurostat 2010) which serves as a strong basis for assessing the multiplier effects that different sectors have on the economy.

Through the debate on the effect that fiscal policy has on gross domestic product and the increasing tendency to have the best possible correlation between government expenditure categories and economic activities, this study aims to bring an alternative approach to assessing the multiplier effects of fiscal policies, especially on budget restructurings in periods of recession, to help the economy through the downturn as quickly as possible and with as few negative effects as possible. The study addresses the multiplier effects of government spending on the case of Albania detailed under COFOG, attempting to open a new window for researchers to help develop fiscal policies that are as effective as possible.

The primary purpose of this paper is to make an analysis of the effects that budget restructuring brings to the economy seen in terms of economic efficiency, especially in the case of natural disasters or pandemics. Jurion (1978) studied the budgetary effects of a redistribution of government expenditure. For this purpose, he used a simple Leontief input–output model, to show in what direction a change in the composition of government spending must be made if the aim is to increase national income. He concluded that in an open economy, undertaking a certain public spending policy with the purpose to increase national income leads to a more significant budget deficit than another type of policy. In developing countries, budget restructurings occur at least once a year, but also in unforeseen and often unsupported periods of economic impact analysis that these changes will bring. In recent years, Albania has often faced these challenges due to natural disasters such as floods or earthquakes and most recently the 2020 pandemic. This approach to the paper was analyzed through the 2020 budget restructuring in order to recover the country from the situation created by COVID-19 by analyzing the initial budget of 2020 and the one revised in July 2020. Through this comparison it is observed whether this restructuring has had greater multiplier effects on the economy or not.

To achieve the purpose of this paper, the methodology used is based on the input–output analysis. This method is used for estimating the spread effects of changes in the final demand for the product of an industry or sector. Numerous authors have emphasized the

importance of the coefficients derived from the IOT system for making the most effective decisions. Knowing the size of multipliers can help design better fiscal plans (Restrepo 2020). Further, Blanchard and Leigh (2013) concluded that an underestimation of fiscal multipliers at the beginning of crises greatly affects growth forecast errors.

The values of this paper are multiple. First, this paper contributes to the literature by analyzing the multiplier effects of government spending in the case of a small economy such as Albania, with high public debt, with the status of candidate country to join the European Union and in times of COVID-19 crisis. In this regard, estimating the size of short-term government expenditure multipliers can be used to design more appropriate fiscal policies. Second, this paper will naturally enable the comparison of multiplier values with previous studies referring to developing countries. Third, to our knowledge, this paper is among the first studies to use the input–output method in the case of Albania to estimate the size of government expenditure multipliers based on a link between COFOG and economic activity.

Due to COVID-19, the Albanian government was forced to review the state budget in order to minimize its negative effects on the economy. For this purpose, there was an increase in total budget expenditures as well as expenditures by institutions at the central level, by 5.1%, and a restructuring of it in order to increase efficiency in the short term and have a higher multiplier effect in economics. This restructuring turned out to have a higher multiplier effect on output at the level of 1538, or about 0.27% higher than the initial multiplier effect, but on the other hand we have a smaller multiplier effect of value added at 0.924, or about 0.21% less than the multiplier effect of the initial budget.

The following is a review of the literature and the theoretical framework on which this study is based. The article then focuses on an analysis of revenue and expenditure trends as well as the budget deficit for the period 2015–2020 for the case of Albania. The fourth part of the paper presents the methodology and data used to analyze the multiplier effects of government spending. The article concludes with the results regarding the impact of changes in domestic demand in each of the sectors related to government spending and at the same time the impact on all other economic sectors related to civil emergency policies in Albania.

## 2. Literature Review

According to the pessimistic situation created from COVID-19, experts from international institutions such as IMF and the WB, in order to limit the negative impacts of this crisis on the public financial capacity or even that of households, as well as to ensure a fast recovery after the end of the crisis, suggested the necessity of using monetary, fiscal and financial incentives (Chen et al. 2020; Loayza and Pennings 2020). Cherif and Hasanov (2020) in their study, among others, suggest state intervention to increase production.

The existing theoretical framework provides inconsistent results on the impact that public spending has on production. On one side is the framework of neoclassical theories which defends the thesis that expansionary fiscal policies can hinder growth by crowding out the private sector (Dornbusch et al. 2018). On the other side is the Keynesianist view, which we often encounter in university textbooks (Keynes 2017) but also in political debates, which supports the idea that fiscal policy can affect output by supporting aggregate demand, so an expansion of spending stimulates demand for work, thus increasing wages and consumption.

Many recent studies have assessed the effects of government spending on macroeconomic activity by emphasizing the role that incentives measures play in pulling a country out of economic downturn. The findings of this empirical research are also ambiguous. On one hand there are numerous studies advocating that the highest impact of fiscal policy has been observed in the periods of recession (Auerbach and Gorodnichenko 2012, 2013). Other authors such as (Riera-Crichton et al. 2015) confirm that fiscal multipliers are larger during recessions in OECD economies, especially when these phases coincide with countercyclical fiscal policies. Another study that supports the use of fiscal stimulus packages to revive the

economy (Koh 2016) concludes that the impact of fiscal policy is greater during financial crises and during business cycle downturns. Muir and Weber (2013) examined the effects of fiscal policy in Bulgaria during the years 2003–2011, also concluding that the greatest impact of fiscal policy on the economy was during the recession periods. Ramey (2019), in his work, computed three categories of fiscal multipliers—government purchases multipliers, tax rate change multipliers and fiscal multipliers—and concluded that higher spending multipliers during recessions or times of high unemployment are fragile. Furthermore, in his work he states the fact that estimated multipliers are not very different across the various methods for identifying government spending shocks in time series. Papaioannou (2019), proved that fiscal consolidation measures bring negative growth effects in both recession and enlargement in the case of Greece.

On the other hand, many researchers have raised doubts about the effectiveness of increasing government spending on economic recovery in conditions where multipliers are at low levels. Researchers (Owyang et al. 2013) using the local projection method (Jordà 2005) raised the question of whether government expenditure multipliers were greater during periods when resources were limited. In their data they also include historical periods that referred to major wars or even depressions in the US and Canada. The results of their study found no evidence that multipliers were greater during periods of recession or high unemployment. This conclusion was also supported in the study of Ramey and Zubairy (2018) who also found no evidence of significant multiplier changes in the US economy in the economic downturn situation which was measured by the unemployment rate.

Meanwhile, Blanchard and Leigh (2013) express in their work that there is no single multiplier for all periods; they can increase or decrease and are diverse for different countries. In the literature regarding the size of fiscal multipliers there are numerous studies that prove a variance of them from negative to four depending on the different methodologies used or even by countries (Riera-Crichton et al. 2015; Batini et al. 2014).

Additionally, Ramey (2019) has made a detailed analysis of different models to evaluate the fiscal multipliers, which he has divided them into three main groups: aggregate country-level time series or panel estimates; estimated or calibrated New Keynesian dynamic stochastic general equilibrium (DSGE) models; and subnational geographic cross-section or panel estimates. The first two models, which can be used at country level, could take values lower or higher than one. He expresses that for government spendings multipliers, most of the estimates are around one or below. The evidence from developed countries suggests that they are positive but less than or equal to unity, meaning that government purchases raise GDP but do not stimulate additional private activity and may actually crowd it out. Regarding the public investment multipliers, these ranged between 0.4 in short run to 1.6 in the long run.

Fiscal multipliers are sensitive to a wide range of characteristics including exchange rate regime, level of economic development, fiscal stance and rate of opening (Ilzetzki et al. 2013). High multipliers for specified sectors could be the result of lower taxes paid directly by producers on those activities and a low denominator value (Hodžič et al. 2018, p. 14). Moreover, Parker (2011) emphases that the size of the multiplier depends on the model chosen.

Multipliers tend to be smaller in developing countries with flexible exchange rate regimes, and with high debt levels (Ilzetzki et al. 2013). Gechert and Will (2012) analyzed 89 papers from 1992 to 2012, with 749 observations of multiplier values analyzing specific characteristics. In their model, the significant variables (characteristics) were related to interest rate, openness to trade and agent behavior. Additionally, in their work they concluded that public investment is the most effective fiscal impulse. This result demonstrates the importance of better public expenditure management, which is usually lacking in developing countries, and a need to design a sustainable fiscal framework referring to public debt (Koh 2016).

In a study conducted for the period 1970–2010 with a large sample of 102 developing countries, where loans were an important source of funding for government spending, the one-year expenditure multiplier turned out to be on average around 0.4 (Kraay 2014). When the multiplier takes values between 0 and 1 it means that an increase by one monetary unit in government spending may result in an increase in output of less than one monetary unit. This author also concluded that multipliers are greater in recession periods, in countries less exposed to international trade, and in countries with flexible exchange rate regimes (Kraay 2012, 2014). The study, referring to Central and Eastern European countries, found a significant multiplier in countries with fixed or volatile regimes, in less developed countries, where public debt ratios relative to GDP were relatively low, and in less open countries referring to international trade (Combes et al. 2015). In the case of Albania, there is a poor empirical literature on the multiplicative effects of fiscal policies and only the study of (Mançellari 2011) shows that multipliers are low. This ambiguity in the findings of international empirical studies according to (Marglin and Spiegler 2013) is attributed to several factors such as: raising different assumptions, using different methodologies, the fact that each study refers to a certain time frame and type of expenditure government.

Previous studies on fiscal incentives in their assessment have mainly used the Vector Auto Regression (VAR) models methodology (Blanchard and Perotti 2002) and structural Vector Auto Regression (SVAR) models (Čapek and Crespo Cuaresma 2020), which are based on time series data.

Regarding the use of multipliers derived from the Input–output table system, there is little evidence, which is based on the final demand components of the Supply–Use components. One of the main works in this regard is that of (Pusch and Rannenberg 2011), who used the input–output system to calculate multipliers for different expenditures categories based on expenditure approach, which is composed as GDP = C + I + G + Nx. In their work, the authors compute the changes for three subcategories: government consumption, government construction expenditure and welfare expenditure (based on consumption quota). Another study based on input–output tables to estimate the multipliers of fiscal policies was that of Pusch (2012). In his work, he split the impact of imports and re-exported goods in the estimation of multipliers for each component of final demand coming from input–output tables, such as public consumption, investment and exports. This study, differing from the above works, takes into consideration the economic purpose of government expenditure by unit. In this sense, for example, the expenditure by the ministry of education is separated to provide expenditure relating to economic activities of public administration, education, construction, research and development, etc.

Furthermore, there are very few case studies both nationally and internationally on the effects that the budget review brings to the economy given the multiplier effect of government spending. This is also since there has been a lack of a comprehensive framework for the correlation between the categories of government expenditures and economic activities. Lately, in the framework of a compilation of government finance statistics (GFS) in the European Union and data dissemination from Eurostat, a Manual to integrate the COFOG (Eurostat 2019) with other classifications has been developed, including NACE (nomenclature statistique des activités économiques). In the framework of reporting on standardized government statistics, EU countries are implementing national methodologies to link the treasury system with the international COFOG classification and national accounts standards according to European System of Accounts (2010).

The methodology of this paper is also focused on the continuation of these methodological standardizations between countries. The first step is to transfer treasury data classified according to COFOG to economic activities. The second one is to analyze the effects of budget restructuring within one year, investigating the effect of fiscal policy on the economy, using the multipliers derived from the input–output table. The aim of this study is to enable the most effective use of treasury data in fiscal policy decisions in the short term, especially in mid-year budget restructuring. This study is based on the data of the year 2020 for Albania. It has a greater importance in situations of economic crisis, such

as that caused by COVID-19, where budget restructuring to minimize negative effects but also to accelerate recovery is seen as a necessary measure to be taken by the policymakers in Albania and beyond.

Basically, all studies conducted in developing countries and low-income countries conclude that fiscal multipliers are low. A study conducted by Honda et al. (2020) showed that the fiscal multipliers in low-income countries are less than half of those in advanced countries. In developing countries, in the absence of sufficient fiscal space (small multipliers), the primary suggestion coming from the WB experts Loayza and Pennings (2020) in terms of COVID-19 is to support the most vulnerable groups, without interrupting public and health care services. It is known that health care expenditures are important determinants of economic growth and the increase in health care expenditures will be associated with a higher economic growth rate (Heshmati 2001). However, in a situation with limited funds, high levels of debt and the immediate blockage of the private sector, using the budget in the most efficient way possible is seen as an essential path to recovery.

## 3. Methodology

This paper aims to estimate the multiplicative impact of government expenditure in output and value added as specified in the national accounts' standards. The methodology is based on the multipliers generated from the input–output framework and the classification of the functions of government, COFOG. To enable the estimation of the government expenditure multipliers, all central government expenditures were taken according to COFOG for 2020 and then they were transferred according to economic activities. In the absence of a standard international methodology, most countries have developed national methodologies to enable this transformation as well as to produce Government Financial Statistics. The EU countries, in the framework of compilation of government finance statistics (GFS), developed a manual to integrate the COFOG (Eurostat 2019) with other classifications, including even that of economic activities (NACE).

In the case of Albania, there have been several attempts to produce GFS, but their publication has not yet been achieved. Under these conditions a logical linking between COFOG and NACE was used to enable this transition, which was analyzed in five digits for COFOG and four digits for NACE to conclude in a linking that is as accurate as possible. This transformation was made in order to know in which economic activity is allocated the money from the state budget and then to use the multipliers coming from the system of input–output tables. For example, if we take into consideration the expenditures from the Ministry of Agriculture, those expenditures could go into different economic activities such as agriculture, construction, transport, public administration, etc. In more general way we can write:

$$EX_i = \sum_{j=n}^{j=1} \text{NACE}_j \tag{1}$$

where $EX_i$ is the government expenses for unit $i$ and $j$ are the economic activities where this expenditure goes to.

In our study, the linking of COFOG to NACE is done with the initial budget data for 2020 and the revised data after the measures for COVID-19 took place in the country. In Table 1, the results of the expenditure of central government institutions by economic activities for both the initial budget of 2020 and the revised one are given.

In the next step, these data are included in the input–output framework to estimate the total economic impact of government expenditure. The input–output analysis is one of the main methods used for estimating the spread effects of changes in the final demand for the product of an industry or sector and input–output tables can also be used for endogenous modeling (Chen et al. 2016). Input–output modelling is not particularly well suited to estimating very large-scale changes to the economy or aspects of the economy experiencing significant or rapid changes from the reference year (Leonard and Looney 2018, p. 7). The IOT framework was created by Wassily Leontief (1986) and found many applications in following upcoming years such as in the work done by Miller and Blair (2022), Fleissner

et al. (1993), Holub and Schnabl (1994), Kurz et al. (1998) and ten Raa (2006). Because of its application at different geographical levels such as country or regional level and the possibilities for comparisons it has become an international standard method used even in the National Accounts in all countries. This framework has been developed from Eurostat and was described in the European System of Accounts (Eurostat 2010) and later even in dedicated of Manual of Supply Use and Input–output Tables (Beutel 2008).

**Table 1.** Total Expenditure of Central Government Institutions by Economic Activities.

| Nace Rev 2 | | | |
|---|---|---|---|
| **Section** | **Description** | **Initial Budget** | **Revised Budget** |
| A | Agriculture, forestry, and fishing | 1167 | 548 |
| C | Manufacturing | 780 | 806 |
| E | Water supply; sewerage, waste management and remediation activities | 8755 | 9660 |
| F | Construction | 3611 | 4125 |
| H | Transportation and storage | 24,503 | 29,710 |
| J | Information and communication | 176 | 182 |
| M | Professional, scientific, and technical activities | 1274 | 1278 |
| N | Administrative and support service activities | 5171 | 5271 |
| O | Public administration and defense; compulsory social security | 144,848 | 149,643 |
| P | Education | 44,716 | 44,141 |
| Q | Human health and social work activities | 79,632 | 85,668 |
| R | Arts, entertainment, and recreation | 2262 | 2174 |
| S | Other service activities | 131 | 131 |
| | Total | 317,025 | 333,337 |

Prepared by the author based on the Ministry of Finance and Economy data for year 2020.

Regarding Albania specifically, the SIOT were produced for the first time in 2015. Liço (2018) mentions the importance of IOT for the analysis of the inter-sectorial linkages. In his study, he identified key sectors in the Albanian economy applying traditional backward and forward linkage methods developed by Chenery and Watanabe.

The economic system to which input–output analysis is applied may be used at micro and macro level. It can be adapted for an enterprise but at the same time for the whole economy as the approach is essentially the same. The structure of each economic sector is represented by appropriate structural coefficients that describe in quantitative terms the relationships between the inputs it absorbs and the output it produces.

The input–output framework consists in three types of tables: supply tables, use tables and symmetric input–output tables, which are the basis for input–output analysis. The SIOT is the base for producing multipliers, which studies the effect of changes in final use on output and related aspects of the economy. The effect on the economy is in three different dimensions.

- Direct impact—the immediate effect caused in the production (output) because of the changes in final demand.
- Indirect impact—the impact caused to other economic activities related to the main activity. This is mostly because of the suppliers because more production needs more inputs.
- Induced impact—related to the income increase of the household in the form of compensation of employments, which may cause increase in spending for consumption and further final goods.

There are different types of multipliers flowing from the input–output system which can be used for different purposes. In regard to economic impact, three of the most frequently used types of multipliers are as described below:

- Output multiplier—measure the effect of one unit change in the final demand of a specific product to the total production in the economy, direct and indirect one.

- Value added multiplier—expressed as the ratio of the direct and indirect GVA changes to the direct GVA change, due to a unit increase in final demand.
- Employment Multipliers—measures the effect on employment of one unit change in the final demand.

As the main reference to produce input–output tables are economic activities based on the statistical classification of economic activities in the European Community (NACE) and Statistical Classification of Products by Activity (CPA); this study has taken into consideration the activities that are mostly related to government expenditure.

As a structure to make the analysis in this study, we used the supply–use tables of 2016, which is the latest available assessment for Albania. Based on the multiplier estimated, it is possible to measure the impact of changes to the domestic demand in each of the sectors related to the government spending and at the same time even the impact in all other economic sectors.

## 4. Results

Since the purpose of this paper is to analyze the multiplier effect on the economy of the state budget and especially the effects of the budget review in times of economic crisis, the initial budget for 2020 is taken into account, which was not affected by the crisis caused due to the situation created by COVID-19 and the revised budget in the second half of 2020, adopted on 2 July 2020, which underwent substantial changes due to COVID-19.

This analysis aims to contribute to an alternative approach for policy makers, for the most effective ways in preparing the budget review considering economic optimization in terms of limited resources. The aim of this paper is to make an analysis of the effects that budget restructuring has brought to the country's recovery in the COVID-19 situation, given the multiplier effects that government spending has had on the economy before and after the pandemic.

### 4.1. Government Expenditure Overview in Albania

Fiscal policies are essential for economic recovery, but they must be carefully formed so as to not create high debt especially in times of recession, even though we live in a time of low interest rates and debt is often treated as a reduced problem (Badia et al. 2020). It is very important that in periods of sustainable growth, concrete measures are taken to have the lowest possible budget deficit and debt reduction to leave opportunities for periods of recession created by various shocks, such as that of the 2019 earthquake where the damage caused based on the Albania Post-Disaster Needs Assessment was estimated at around EUR 844 million. These measures should be considered especially for small and developing countries such as Albania. In this part of the paper, an analysis has been made for the period 2015–2020, to see the trends of revenues and expenditures and the budget deficit during the last five years to address the pre-crisis budget situation of 2019 and 2020.

Since the effects of the financial crisis of 2008 in Albania reached their maximum induced effect in 2013 where the country's economy achieved the lowest growth by only 1%, the five-year period 2015–2019 should have been a period of fiscal sustainability and return of growth to pre-crisis rates. What can be seen from the analysis, budget expenditures in the country have had an average level of 8.1% higher than budget revenues. During all these years, as shown in Figure 1, the budget of the Albanian government has been characterized by higher expenditures than revenues, which has been at the minimum level in 2018 with 5.8% more.

In terms of the 2020 budget, considering the damage caused by the 2019 earthquake, it was projected that expenditures would be 7.8% higher than revenues, which were also projected to have a considerable increase. The emergence of COVID-19 in March and the effects of the closure forced the government to review the budget in the second half of 2020. The budget, on the one hand, witnessed a total 12.4% decline in revenues but, on the other hand, displayed a faster recovery effect on the economy; furthermore, there was no reduction of expenditures but an increase of 5.5%. This difference was covered by the

increase of the budget deficit which in the budget review, compared to the initial one, saw an increase of 235%.

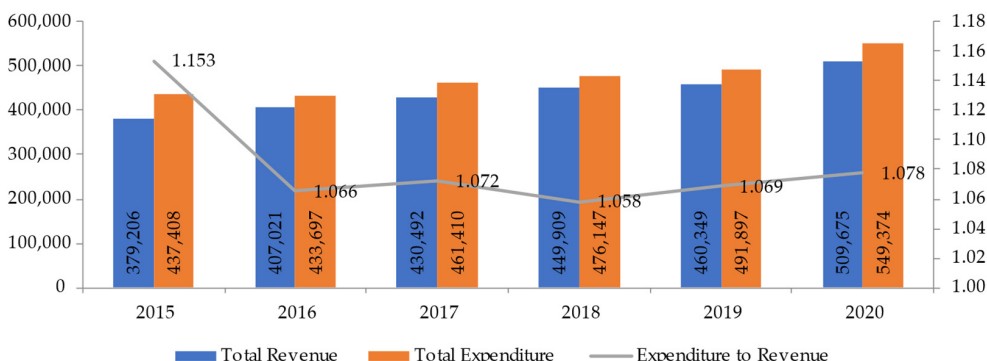

**Figure 1.** Total Budget 2015–2020 (million lek). Source: Prepared by the author based on the Ministry of Finance and Economy data for year 2020.

Since this paper is focused only on the category of expenditures by budgetary institutions, in Figure 2 a detailed analysis of this component is made according to the main categories based on COFOG. What stands out is the fact that none of the functions has had a continuous increase over the years, which is an indication of the non-existence of long-term structural policies in the country.

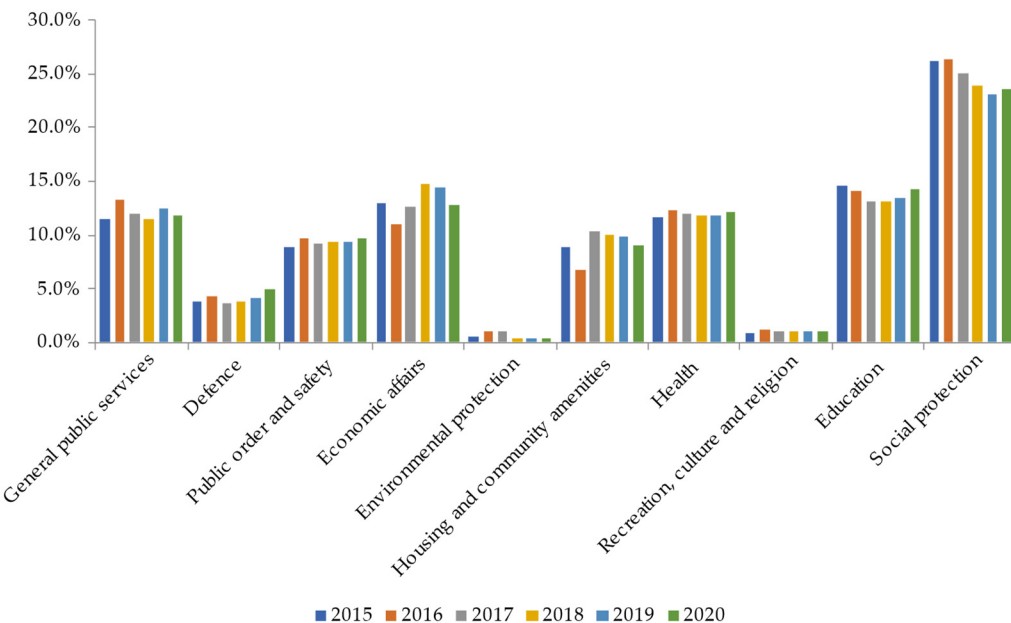

**Figure 2.** Central Government Institution Expenditure Structure by COFOG for 2015–2020. Source: Prepared by the author based on the Ministry of Finance and Economy data for year 2020.

Regarding the situation created after the budget review in 2020, as presented in Figure 3, it is noted that there was an increase in total expenditures for budgetary institutions of 5.1% while total expenditures increased by 5.5%. The restructuring was carried out by giving priority to functions that were more directly related to the effects caused by COVID-19. More specifically, the restructuring was accompanied by an increase of 15.9% on housing, 11.2% on social protection expenditures and 10.6% on economic issues, while there was a reduction in defense spending of 11.8% or even in environmental issues of 6.3%.

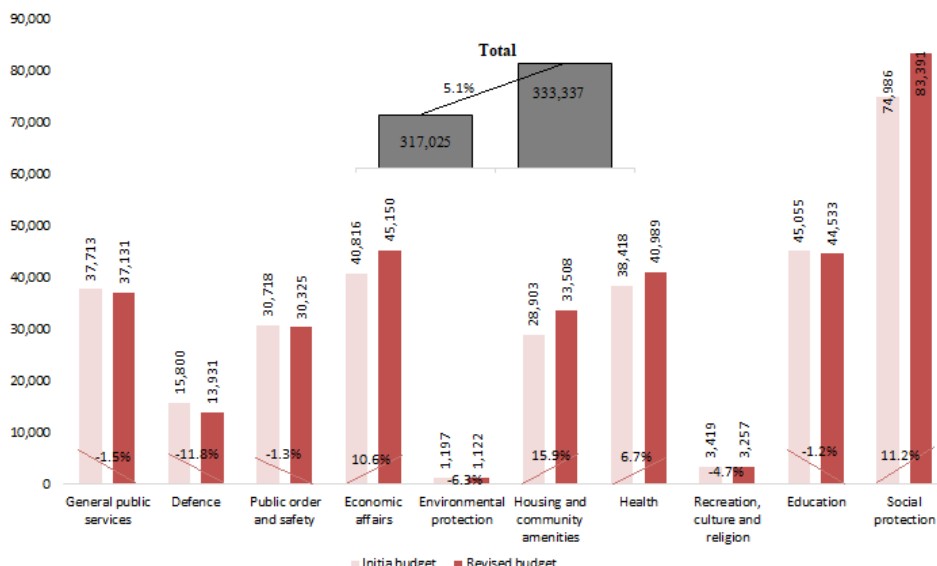

**Figure 3.** Central Government Institution Expenditure by COFOG for 2020 (million lek). Source: Prepared by the author based on the Ministry of Finance and Economy data for year 2020.

*4.2. Government Budget Review and Its Economic Impact*

The purpose of this study is to create a framework, which made it possible to assess the effects on the economy that come due to the distribution of expenditures by economic activities. This helps in decision-making for budget restructuring based on the principle of economic efficiency not only in periods of recession but also in budget planning by giving priority to expenditures which have greater effects on the economy.

COVID-19 pressured many countries to redistribute their initial budget plan and give a specific focus to the main sectors fighting the pandemic. Cakmakli et al. (2020) has used I–O linkages with the other domestic and international sectors to estimate the external financing needs because of the COVID-19 situation. In their model they take into account the effects of the pandemic on domestic demand and foreign demand and also relate capital flows and trade in an emerging market. Through this model they show that sectors with stronger I–O links suffer from larger COVID-19-related losses and sectors that finance these stronger production links through capital flows suffer even more.

Referring to the initial data of 2020, the level of expenditure by central institutions was predicted to be in the amount of ALL 317,025 million, which would have produced an output value in the economy of ALL 486,201 million, or 1534 times more than the initial expenditures, and an added value of ALL 293,561 million or 0.926 times more than the initial expenses.

Due to COVID-19, the Albanian government was forced to review the state budget to minimize the negative effects caused to the economy. In order to achieve this purpose, there was an increase in total budget expenditures as well as expenditures by institutions at the central level, by 5.1% and a restructuring of it to increase efficiency in the short term and have a higher multiplier effect in economics.

This restructuring had a higher multiplier effect on output at the level of 1538 or about 0.27% higher than the initial multiplier effect; but, there was a smaller multiplier effect at 0.924 or about 0.21% less than the multiplier effect of the initial budget for value added.

Table 2 provides a summary of these results while a detailed multiplier information for each economic activity and the effects it has brought is given in the table of Appendix A.

**Table 2.** Output and added value multiplicative effects of government expenditure.

| Economic Activity | Shock | | Total Output Change | | Change in Value Added (GDP) | |
|---|---|---|---|---|---|---|
| | Initial | Revised | Initial | Revised | Initial | Revised |
| Total Impact | 317,025 | 333,337 | 486,201 | 512,587 | 293,561 | 308,007 |
| Multiplier | | | 1.534 | 1.538 | 0.926 | 0.924 |
| Change | | | 53.40% | 53.80% | −7.40% | −7.60% |

Source: Prepared by the author based on the Ministry of Finance and Economy data (2020) and input output tables from INSTAT (2016).

## 5. Discussion

The mid-year budget review is a well-known phenomenon in developing countries. However, despite the major changes that can be considered, analyzing the multiplier effects on the economy can minimize the budget restructuring effects. Since there is a lack of specific papers regarding the economic effects of midyear budget reallocation, this study brings into focus specific topics that can be subject of further analysis such as:

- There is no standardized link between COFOG and NACE, as this link is made manually for each code by the authors themselves. In these circumstances it is worth mentioning the margin of judgment error, however the structure used for the initial and revised data is the same, thus eliminating structural errors.
- The study is based only on expenditure data of central institutions as these data are available according to COFOG. In the future, this analysis will have to be extended to all components of the budget, including local government, to have a more inclusive assessment.
- The study treats total expenditures undivided into their constituent components by activities, which is based on the assumption that all components have the same multiplier effect on economic activity.
- Further studies could be done in the future dividing the government spendings by their main components by expenditure approach and by their purpose.
- This work can be followed with parallel analyses for other countries based on their level of development.
- More studies have to be done to catch the impact in the short run of the investments and government expenditures.

## 6. Conclusions

The paper aims to bring some conclusions regarding the importance of analyzing the multiplicative effect of budget reallocation on the economy. In this regard, some conclusions that come out of this study are as follows:

- The state budget in recent years has had a deteriorating trend, making it difficult to undertake fiscal policies that are as aggressive as possible in order to exit the crisis caused by the earthquakes of 2019 and to have the most comprehensive support in the presence of COVID-19.
- Budget restructuring in Albania has brought an improvement of the multiplier effect on production of 0.27%.
- What results from the study is that for the added value there has been a deterioration of the total multiplier effect from 0.926 to 0.924. These results stress the need for more in-depth analysis of the effects of fiscal policies on the economy during budget reviews, so that the multiplier effects of spending are as high as possible in the country.

**Author Contributions:** Conceptualization, L.S. and P.G.; methodology, L.S.; software, L.S.; validation, L.S. and P.G.; formal analysis, L.S.; investigation, L.S.; resources, L.S. and P.G.; data curation, L.S.; writing—original draft preparation, L.S. and P.G.; writing—review and editing, L.S. and P.G.; visualization, L.S. and P.G.; supervision, L.S. and P.G.; project administration, and P.G.; funding acquisition, L.S. and P.G. All authors have read and agreed to the published version of the manuscript.

**Funding:** This research received no external funding.

**Institutional Review Board Statement:** Not applicable.

**Informed Consent Statement:** Not applicable.

**Data Availability Statement:** https://financa.gov.al/buxheti-2/; https://financa.gov.al/buxheti-2020/; https://financa.gov.al/cash-accrual-2020/; http://www.instat.gov.al/en/themes/economy-and-finance/supply-use-and-input-output-tables/ (accessed on 10 March 2023).

**Conflicts of Interest:** The authors declare no conflict of interest.

## Appendix A. Impact of Government Expenditures in Total Output and Added Value

| No | Economic Activity | Shock | | Total Output Change | | Change in Value Added (GDP) | |
|---|---|---|---|---|---|---|---|
| | | Initial | Revised | Initial | Revised | Initial | Revised |
| 1 | Agriculture, forestry, and fishing | 1167 | 548 | 11,629 | 11,352 | 8346 | 8147 |
| 2 | Mining and quarrying | 0 | 0 | 4007 | 4276 | 2098 | 2239 |
| 3 | Manufacture of food products, beverages, and tobacco products | 0 | 0 | 3105 | 3244 | 801 | 837 |
| 4 | Manufacture of textiles, wearing apparel and leather products | 0 | 0 | 795 | 841 | 387 | 409 |
| 5 | Manufacture of wood and paper products, and printing | 780 | 806 | 7238 | 7595 | 2768 | 2904 |
| 6 | Manufacture of coke and refined petroleum products | 0 | 0 | 1468 | 1185 | 254 | 205 |
| 7 | Manufacture of chemical and pharmaceutical products | 0 | 0 | 11,537 | 12,320 | 4523 | 4830 |
| 8 | Manufacture of rubber and plastic products and other non-metallic mineral products | 0 | 0 | 6514 | 7009 | 1736 | 1868 |
| 9 | Manufacture of basic metals and fabricated metal products, except machinery and equipment | 0 | 0 | 11,953 | 12,755 | 2878 | 3071 |
| 10 | Manufacture of machinery and equipment | 0 | 0 | 6076 | 6496 | 2610 | 2790 |
| 11 | Manufacture of furniture; other manufacturing; repair and installation of machinery and equipment | 0 | 0 | 3658 | 3907 | 1347 | 1439 |
| 12 | Electricity, gas, steam and air-conditioning supply | 0 | 0 | 5037 | 5339 | 4050 | 4293 |
| 13 | Water supply | 8755 | 9660 | 11,250 | 12,291 | 5789 | 6325 |
| 14 | Sewerage, waste management and remediation activities | 0 | 0 | 1675 | 1785 | 671 | 715 |
| 15 | Construction | 3611 | 4125 | 24,205 | 26,246 | 7678 | 8325 |
| 16 | Wholesale and retail trade and repair of motor vehicles and motorcycles | 0 | 0 | 2179 | 2352 | 1443 | 1558 |
| 17 | Wholesale trade, except of motor vehicles and motorcycles | 0 | 0 | 7855 | 8361 | 5111 | 5440 |
| 18 | Retail trade, except of motor vehicles and motorcycles | 0 | 0 | 3393 | 3575 | 2328 | 2453 |
| 19 | Land transport and transport via pipelines | 0 | 0 | 5289 | 5591 | 1855 | 1961 |
| 20 | Water and air transport; warehousing | 24,503 | 29,710 | 28,425 | 34,009 | 15,506 | 18,553 |
| 21 | Postal and courier activities | 0 | 0 | 1550 | 1676 | 900 | 973 |
| 22 | Accommodation and food service activities | 0 | 0 | 5241 | 5518 | 2383 | 2509 |
| 23 | Publishing, audiovisual and broadcasting activities | 176 | 182 | 6199 | 6428 | 2543 | 2638 |
| 24 | Telecommunications | 0 | 0 | 4606 | 4849 | 1552 | 1634 |
| 25 | Computer programming, consultancy, and related activities; information service activities | 0 | 0 | 2828 | 2948 | 1238 | 1290 |
| 26 | Financial and insurance activities | 0 | 0 | 6610 | 7009 | 3993 | 4233 |
| 27 | Real estate activities | 0 | 0 | 1619 | 1728 | 1375 | 1468 |
| 28 | Legal and accounting activities; management consultancy activities; architectural and engineering activities | 0 | 0 | 5563 | 6020 | 2577 | 2788 |
| 29 | Scientific research and development; other professional, scientific, and technical activities | 1274 | 1278 | 2798 | 2909 | 1145 | 1191 |
| 30 | Administrative and support service activities | 5171 | 5271 | 12,623 | 13,254 | 6906 | 7251 |
| 31 | Public administration and defense; compulsory social security | 144,848 | 149,643 | 145,369 | 150,190 | 103,424 | 106,854 |
| 32 | Education | 44,716 | 44,141 | 46,902 | 46,408 | 38,410 | 38,005 |
| 33 | Human health activities | 79,632 | 85,668 | 80,444 | 86,505 | 51,338 | 55,207 |
| 34 | Arts, entertainment, and recreation | 2262 | 2174 | 3696 | 3656 | 2566 | 2538 |
| 35 | Other services and activities of households | 131 | 131 | 2864 | 2958 | 1032 | 1066 |
| | Total Impact | 317,025 | 333,337 | 486,201 | 512,587 | 293,561 | 308,007 |

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
