# Peer review of "Economic Role of Government Budget Revision in the Presence of COVID-19"

_economies, doi:10.3390/economies11040118_

Round 1
Reviewer 1 Report
Like many countries around the world, Albania has been affected by the COVID-19 pandemic, which has had significant impacts on the country's economy. In response to the pandemic, the Albanian government has implemented a range of measures aimed at mitigating the economic impact and supporting businesses and individuals affected by the crisis. The author(s) of the paper use the Albanian government's budget changes to empirically study the effects of government spending.
In macroeconomics, a fiscal multiplier is a measure of how much a change in government spending or taxation affects a country's national income or gross domestic product (GDP). It is a key concept in fiscal policy and is used to assess the effectiveness of government spending and tax policies. Fiscal multipliers can be positive, negative, or zero. A positive multiplier indicates that a change in government spending or taxation has a positive impact on national income or GDP. A negative multiplier indicates the opposite, while a multiplier of zero indicates that there is no effect on national income or GDP. Ramey (2019) clearly outlines the theoretical implications of the various macroeconomic models.
The size of the fiscal multiplier depends on several factors, such as the type of spending or taxation being implemented, the size of the government's budget deficit, and the overall state of the economy. For example, during times of recession or economic downturns, fiscal multipliers tend to be larger, as additional government spending can stimulate demand and help boost economic activity. There are different types of fiscal multipliers, including the government spending multiplier and the tax multiplier. The government spending multiplier measures the impact of changes in government spending on national income or GDP, while the tax multiplier measures the impact of changes in taxation. While it is true that the empirical literature on tax multipliers does not converge on unambiguous results. Nevertheless, there are meta-analyses analyzing the findings already obtained Gechert and Will (2012),
Gechert and Rannenberg (2018), Capek, Crespo and Cuaresma (2020). This article focuses on the estimation of these multipliers. Since the literature on the subject uses imprecise vocabulary, it would be useful to specify whether the authors want to estimate fiscal multipliers or expenditure multipliers. The authors could be more explicit about their methodology. It seems to me that their approach is similar to Pusch (2012) or Pusch and Rannenberg (2011). A quick exposition of the estimated equations would be welcome.
Čapek, J., & Crespo Cuaresma, J. (2020). We just estimated twenty million fiscal multipliers. Oxford Bulletin of Economics and Statistics, 82(3), 483-502.
Gechert, S., & Will, H. (2012). Fiscal multipliers: a meta-analysis of the literature. Chemnitz University of Technology-Working paper.
Gechert, S., & Rannenberg, A. (2018). Which fiscal multipliers are regime‐dependent? A Meta‐regression analysis. Journal of Economic Surveys, 32(4), 1160-1182.
Pusch, T. (2012). Fiscal spending multiplier calculations based on input-output tables–an application to EU member states. European Journal of Economics and Economic Policies: Intervention, 9(1), 129-144.
Pusch, T., & Rannenberg, A. (2011). Fiscal spending multiplier calculations based on Input-Output tables–with an application to EU members (No. 1/2011). IWH Discussion Papers.
Ramey, V. A. (2019). Ten years after the financial crisis: What have we learned from the renaissance in fiscal research?. Journal of Economic Perspectives, 33(2), 89-114.
Reviewer 2 Report
Dear Authors, this paper is very well written. I like the paper and thank you for the opportunity to read yours work.
This article is of extremely high scholarly standards, it is original in numerous respects and certainly most timely in terms of Economies relevance. Key area of improvements are editorial and contextual. By point of view:
- editorial - using less complex sentence structures and clarifying specialist terms would make the article more reader friendly.
- issues for further work - the conclusions should draw the essence of the study from the propositions made and recommend further direction for future research directions.
